# Clinical profile and hospital outcomes of children with Down syndrome diagnosed with congenital heart disease in a developing country: A retrospective study

Zawadi Edward Kalezi[1,2]*, Jimmy France Mchalange[1], Naizihijwa Gadi Majani[1,2], Alphonce Nsabi Simbila[3,4], Vivienne Aiyana Mlawi[1], Deogratias Arnold Nkya[1,2], Peter Richard Kisenge[1]

**1** Department of Cardiology, Jakaya Kikwete Cardiac Institute, Dar es Salaam, Tanzania, **2** Department of Paediatrics and Child Health, Muhimbili University of Health and Allied Sciences, Dar es Salaam, Tanzania, **3** Emergency Medicine Department, Muhimbili University of Health and Allied Sciences, Dar es Salaam, Tanzania, **4** Emergency Medicine Department, Muhimbili National Hospital, Dar es Salaam, Tanzania

* ezawadi8@gmail.com

## Abstract

### Background

The World health organization (WHO) identifies Down syndrome (DS) as one among common congenital disorders, along with congenital heart defects. Approximately fifty percent of children diagnosed with DS are affected by congenital heart defects which significantly impact their survival. Nevertheless, in low and middle income countries, there is limited published data on congenital heart defects in children with DS and their treatment outcomes. Therefore, this study aimed to document the clinical characteristics and hospital outcomes of children with DS who were diagnosed with heart defects. The goal is to assist healthcare providers and policymakers in improving care for these children.

### Methods

This was a retrospective descriptive study of children with DS diagnosed with heart disease admitted at the Jakaya Kikwete Cardiac Institute (JKCI) from December 2022 through December 2024. Socio-demographics, clinical characteristics, and survival data were extracted from medical records. Frequencies and proportions were calculated for categorical variables. The description of mean with standard deviation (SD) and median with and interquartile range (IQR) were calculated for continuous data. For the missing data, the case deletion approach was used.

### Results

In two years, out of 1,356 admitted children, data from 104 children with Down syndrome were analysed. Most of them, 93.2%, were aged below 5 years, with a

**Data availability statement:** All relevant data are within the paper.

**Funding:** The author(s) received no specific funding for this work.

**Competing interests:** The authors have declared that no competing interest.

**Abbreviations:** AVSD, Atrioventricular septal defect; CHD, Congenital heart disease; DS, Down syndrome; ECHO, Echocardiography; ICU, Intensive care unit; PDA, Patent ductus arteriosus; TOF, Tetralogy of Fallot; VSD, Ventricular septal defect.

slight predominance of male children, of almost 58%. Fifty percent (58/104) of study participants resided in the coastal region of Tanzania, followed by the northern zone (19.2%, 20/104). The most frequent cardiac diagnoses were AVSD, 46.2% and VSD, 14.4% with a median age at diagnosis of 5 months (IQR, 3.3–10). The median age of the mothers was 38 years (IQR, 38–42) while the mean age of the fathers was 36.9 years (SD, ± 7.1). At discharge, nearly a quarter, 19.2% of children had prolonged hospital stays of more than 2 weeks, and 7.7% (8/104) of enrolled children died. Nevertheless, 4.8% of children with Down syndrome underwent open cardiac surgery during the study period.

## Conclusion

Advanced maternal age, paternal age and AVSD were frequently observed in this study, nevertheless; a significant number of the enrolled children were first diagnosed beyond 5 months of age. Hence, the study recommends early screening echocardiography among children with Down syndrome and more studies with a large sample size to evaluate long-term outcomes in this population.

## Background

Down syndrome (DS) is one of the most common chromosomal disorders with an incidence of approximately 1 in 733 live births [1]. A population-based study in the USA, estimated the live birth prevalence for DS to be approximately 12.6 per 10,000 (1 in 792) in the era of DS-related elective pregnancy terminations [2]. Despite limited studies in developing countries, the prevalence of DS is reported to be higher compared to developed settings. However, this is contrary to what was reported in the 1990s. This could be attributable to many factors including advanced maternal age at birth, elective DS-related pregnancy terminations following prenatal diagnosis, and traditional beliefs [3,4].

Advanced maternal age remains the primary risk factor of having a child with trisomy 21; however, the younger women with higher overall birth rates also increase the risk of DS [1]. Furthermore, age below 20 years and above 40 years has been mentioned to increase the likelihood of chromosomal and non-chromosomal abnormalities. A nine-year cross-sectional survey was conducted after a few studies reported a shift to younger maternal age for DS in the Rwandan population. Conversely, this study found that the mean maternal age at birth of DS was significantly higher compared to the general population [5].

Down syndrome is reported to be associated with congenital anomalies, in addition to cognitive impairment. Around 50% of the children with DS are more prone to have congenital heart defects (CHD) [1,6]. This includes atrioventricular septal defect (AVSD) which has been reported to be the most common type of CHD among the DS population. However, recent studies have shown that the ventricular septal defect (VSD) is also a frequently reported lesion [6,7]. A Tanzanian single-centre study

screened 3982 children identified 1371 to have CHD, and the most common associated syndrome was DS accounting for 12.8% of all recruited children [8].

The life expectancy of children with DS is lower compared with the non-DS population [1]. This finding is attributed to presence of the CHD, and other comorbidities like leukaemia and seizures. The presence of CHD in these children leads to recurrent respiratory infections, congestive heart failure and pulmonary hypertension (PHT). Nevertheless, children with DS have a propensity to develop persistent PHT and irreversible pulmonary vascular disease earlier than in non-DS children. Thus, early repair of heart defect in the first year of life has been recommended to minimize the risks of heart failure, and irreversible pulmonary vascular disease [1,9].

CHD is one among the major predictors of survival in children with DS, however, documentation on its burden and related outcomes is limited in developing settings. We, therefore, conducted this study to demonstrate the clinical profile and hospital outcomes of children with Down syndrome in the hope of aiding healthcare workers and policy makers regarding the care of this special group of children.

## Methods

### Study design and setting

This was a hospital-based retrospective descriptive study conducted among DS children with CHD admitted at the Jakaya Kikwete Cardiac Institute (JKCI) from December 2022 through December 2024. JKCI is a 150-bed capacity national referral and teaching centre located in Dar es Salaam, Tanzania. It provides specialized cardiac services including percutaneous interventions and open cardiac surgery for both children and adults.

### Study population and participants

All DS children with CHD aged 1 month to 18 years admitted to the paediatric unit during the study period were enrolled. However, participants with incomplete data and those readmitted were excluded during chart review. The patient data were accessed for research purposes from 23rd June 2025.

### Study variables

Medical records of all children with DS at JKCI within 2 years were reviewed. Clinical and sociodemographic parameters were age, gender, paternal age, maternal age, parity, address (residence), date of admission and discharge, history of cardiac surgery/ cardiac catheterization, laboratory findings, documented cardiac and non-cardiac diagnoses, age at cardiac diagnosis and survival (death/ alive) at discharge.

### Data management and analysis

Data entry and cleaning were done using Statistical Package for Social Science (SPSS) version 25. The description of mean with standard deviation (SD) and median with and interquartile range (IQR) were calculated for continuous normally distributed and skewed data respectively. Frequencies and proportions were calculated for categorical variables. For the missing data, the case deletion approach was used.

### Ethical consideration

Ethical approval to conduct the study was obtained from the Ethics Review Committee of the Jakaya Kikwete Cardiac Institute with ethical clearance number AB.123/307/01K/39. All the data were anonymized to ensure patient confidentiality and being retrospective study an informed written consent from each study participant was not needed per ethical committee.

## Results

Over a two-year period, a total of 1,356 children were admitted to the JKCI. Of these, 165 children had DS. Thirty-four children had missing information, and 27 children who were readmitted were excluded. Therefore, 104 study participants were included in the final analysis (Fig 1).

As shown in **Table 1**, Out of 104 children with DS, most of them, 93.2%, were aged below 5 years with a slight predominance of male children of almost 58%. The median child's position in the family was 3 (IQR, 2–5) and the majority

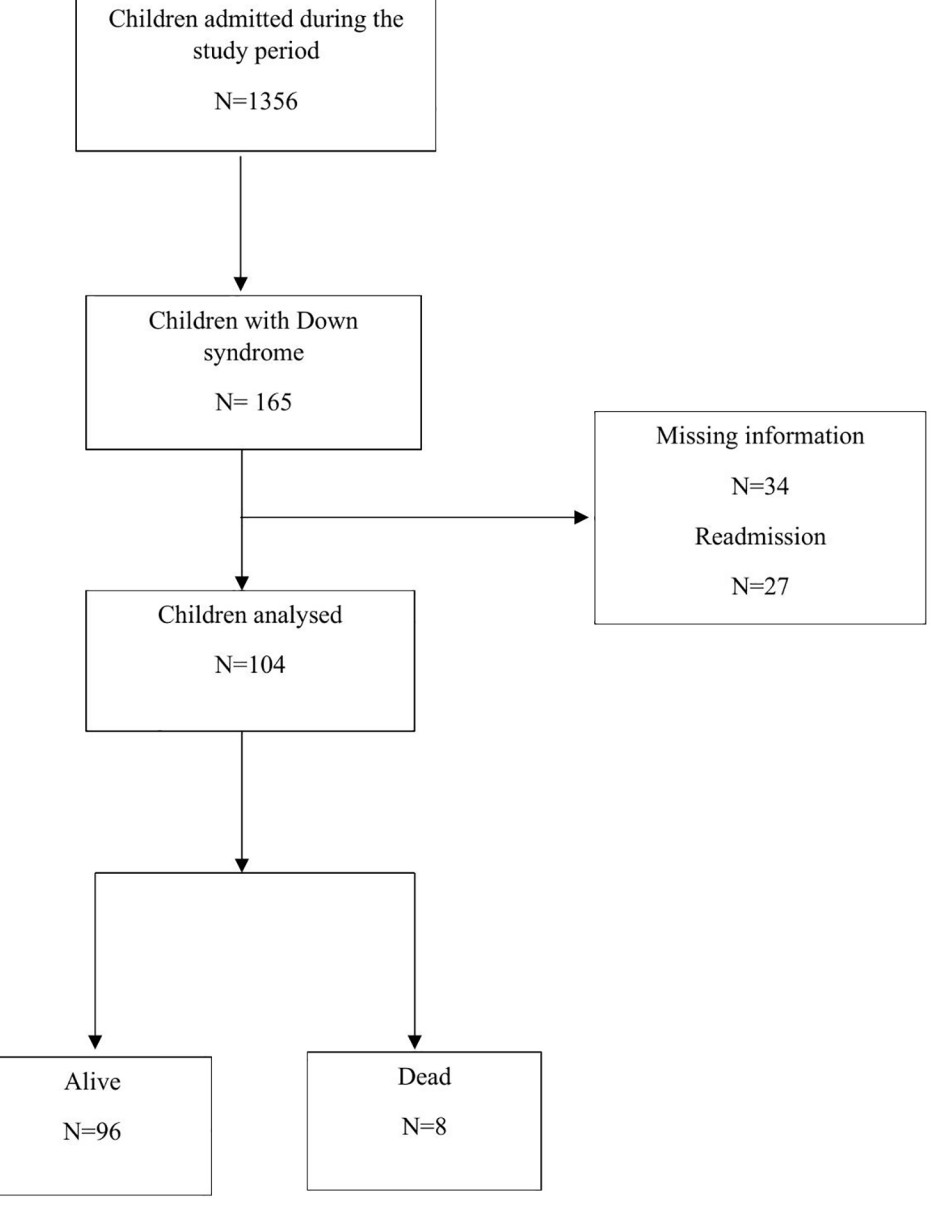

**Fig 1. Flow diagram of study participants and outcomes.** Over a two-year period, 1,356 children were admitted to the JKCI. Of these, 165 children had DS. Thirty-four children had missing information and 27 readmitted children were excluded. A total of 104 participants were included in the final analysis.

**Table 1. Demographic and clinical characteristics of children with Down syndrome admitted at JKCI in Tanzania, 2023-2024 (N = 104).**

| Variable | Category | Frequency (%) |
|---|---|---|
| Age (yr) | < 1 | 54 (51.9) |
| | 1-<5 | 43 (41.3) |
| | 5-<10 | 6 (5.8) |
| | 10-18 | 1 (1.0) |
| Gender | Male | 60 (57.7) |
| Place of residence | Coastal/ Eastern zone | 58 (55.8) |
| | Northern zone | 20 (19.2) |
| | Lake zone | 9 (8.7) |
| | Western zone | 3 (2.9) |
| | Central zone | 2 (1.9) |
| | Southern highlands | 6 (5.8) |
| | Southern zone | 6 (5.8) |
| Parity | ≤ 2 | 29 (27.9) |
| | > 2 | 75 (72.1) |
| | Median (IQR) | 3 (2–5) |
| Maternal age | Median (IQR) | 38 (33-42) |
| Paternal age | Mean ± SD | 36.9 ± 7.1 |
| Anaemia | Yes | 65 (62.5) |
| | Non severe | 61 (58.7) |
| | Severe | 4 (3.8) |
| Underlying cardiac diagnosis | VSD | 15 (14.4) |
| | AVSD | 48 (46.2) |
| | PDA | 7 (6.7) |
| | ASD | 7 (6.7) |
| | TOF | 8 (7.7) |
| | Multiple defects | 17 (16.3) |
| | Structurally Normal heart | 2 (1.9) |
| Age at the time of cardiac diagnosis | Median (IQR) | 5 (3.3-10) |
| Other diagnoses | Heart failure | 46 (44.2) |
| | Pneumonia | 25 (24) |
| | Sepsis | 24 (23.1) |
| | Others | 9 (8.7) |

(75/104) were in third position and above. Fifty percent (58/104) of study participants were residing in the coastal region of Tanzania, followed by the northern zone (19.2%, 20/104). The most frequent cardiac diagnoses were AVSD, 46.2% and VSD, 14.4% with a median age at diagnosis of 5 months (IQR, 3.3–10). Other documented diagnoses were heart failure (44%), pneumonia (24%), and sepsis (23%). The median age of the mothers was 38 years (IQR, 38–42) while the mean age of the fathers was 36.9 (SD, ±7.1).

At discharge, nearly a quarter of the children (19.2%) had prolonged hospital stays of more than two weeks, and 7.7% (8/104) of enrolled children died (Fig 2). Nevertheless, only 4.8% of children with DS underwent open cardiac surgery during the study period.

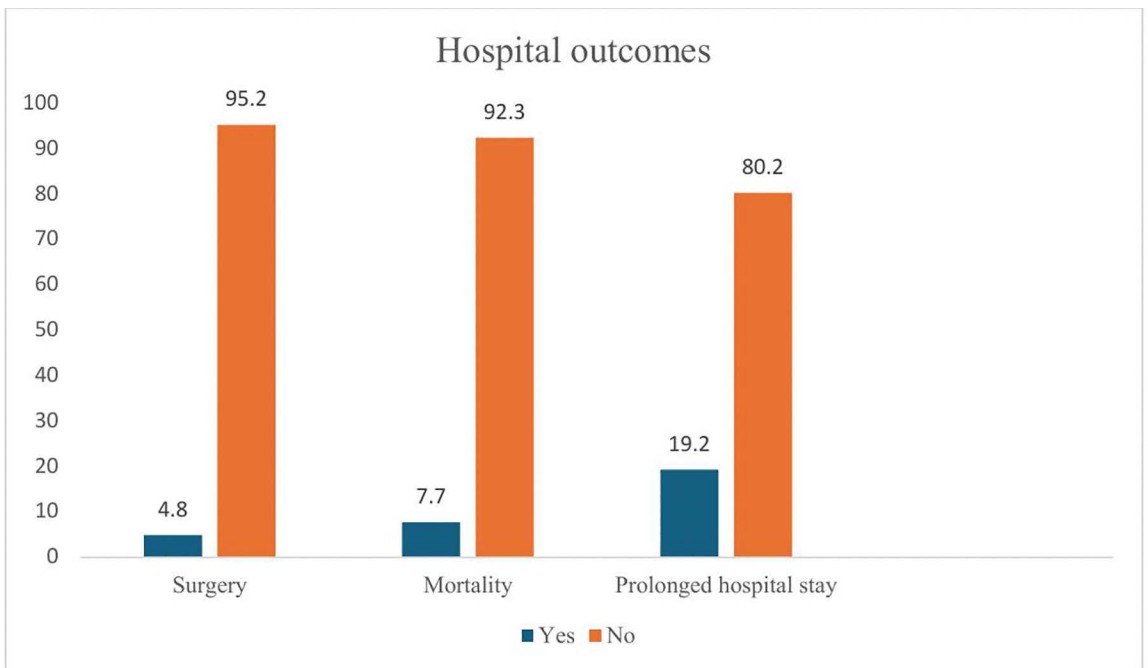

**Fig 2. Hospital outcomes at discharge among enrolled children.** A total of 19.2% of children had prolonged hospital stays of more than two weeks, 7.7% (8/104) died, and 4.8% of children with Down syndrome (DS) underwent open cardiac surgery during the study period.

## Discussion

This study determined the clinical profile and hospital outcomes of the children with Down syndrome admitted at JKCI from December 2022 through December 2024. In 2 years, 165 children with Down syndrome were admitted, however, only 104 children were included in the final analysis due to missing information and readmissions. AVSD and VSD were the most frequent cardiac lesions diagnosed beyond 5 months of age in half of the study participants. The present study indicates that half of the mothers were aged 38 years and above while the average fathers' age was 36.9 years. Nevertheless 4.8% of the children with Down syndrome underwent open cardiac surgery and only eight children died out of 104 enrolled children.

In this study, almost half of the children had AVSD, which is comparable to other previous studies [1,8,10]. However, recent studies conducted in Malaysia and Egypt reported VSD being the commonest cardiac lesion among the DS cohort of children [6,7]. This variability of the cardiac lesions could be due to genetic variation, since both studies are from the Asian population.

Cardiac defects such as AVSD and other CHD with increased pulmonary blood flow need to have early surgical repair to minimize the risk of developing irreversible pulmonary vascular diseases, especially in children with DS [1]. However, in the present study, we observed more than half of children were first diagnosed with CHD later than 5 months of age. This fact increases the odds of these children, either having delayed surgical repair or not fit for surgical intervention, particularly in low-resource settings. There are several reasons for late surgical repair in developing countries, considering the pathway of care, one being diagnosed at an advanced age plus a long waiting surgical list. Nevertheless, in two years, 4.8% of the children with Down syndrome underwent cardiac surgeries.

Advanced maternal age has been reported consistently over the years to be a risk for having a child with DS [1,10]. This study also demonstrates that half of the mothers of these children were aged 38 years and above. Additionally, we

observed that the average age of the fathers was 36.88 years. Whether it is advanced or not, to my knowledge, there are limited studies discussing further the paternal age to be a risk factor for DS. Some studies reported the influence of paternal age on DS, however, it is in association with advanced maternal age [11–13].

We also observed that half of studied children were residing in the coastal region of Tanzania, encompassing five regions including Dar es Salaam. This finding could possibly be the proximity of these areas to our centre; however, this opens up a way for further studies regarding this observation.

This study had some limitations, including being retrospective, and secondly, the small number of study participants, which limited further statistical analysis.

## Conclusion

Down syndrome is the most common syndrome among children with congenital heart disease, and advanced maternal age, and AVSD were frequently observed in this study. Furthermore, half of the enrolled children were first diagnosed at 5 months of age or above. Hence, this study recommends early screening echocardiography among children with Down syndrome during outpatient clinics and further studies with a large sample size to evaluate long-term outcomes in this population.

## Supporting information

**S1 File. Strobe checklist-Down syndrome.**
(DOCX)

## Acknowledgments

We are grateful to all doctors and nurses who were involved in these patients.

## Author contributions

**Conceptualization:** Zawadi Edward Kalezi, Jimmy France Mchalange, Naizihijwa Gadi Majani, Alphonce Nsabi Simbila.

**Writing – original draft:** Zawadi Edward Kalezi.

**Writing – review & editing:** Zawadi Edward Kalezi, Jimmy France Mchalange, Naizihijwa Gadi Majani, Alphonce Nsabi Simbila, Vivienne Aiyana Mlawi, Deogratias Arnold Nkya, Peter Richard Kisenge.

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
