## [Decision Letter · Decision Letter 0]

18 Dec 2025

Dear Dr. Kazeni

Thank you for submitting your manuscript to PLOS ONE. After careful consideration, we feel that it has merit but does not fully meet PLOS ONE’s publication criteria as it currently stands. Therefore, we invite you to submit a revised version of the manuscript that addresses the points raised during the review process.

We look forward to receiving your revised manuscript.

Kind regards,

Palesa Motshabi Chakane, PhD

Academic Editor

PLOS One

2. Please include a separate caption for each figure in your manuscript.

Additional Editor Comments:

Major Revision rlated to data presented. In text.

Reviewers' comments:

Reviewer's Responses to Questions

**Comments to the Author**

1. Is the manuscript technically sound, and do the data support the conclusions?

Reviewer #1: Yes

2. Has the statistical analysis been performed appropriately and rigorously?

Reviewer #1: Yes

3. Have the authors made all data underlying the findings in their manuscript fully available?

Reviewer #1: Yes

4. Is the manuscript presented in an intelligible fashion and written in standard English?

Reviewer #1: No

Reviewer #1: The outline of the aim and objectives is poorly worded.

The presentation of the data is adequate in the results and the discussion section but the abstract is poorly outlined.

There are colloquialisms involved in the discussion that are unnecessary e.g "at the other end of the pyramid"

Why are the mothers' ages described in median and the fathers in mean format.

There could be more referencing in the discussion portions, there are many assertions that appear to be congruent but they could be strengthened by seeking references.

The references are Vancouver style but there are at least 5 references that are in the incorrect format.

The Ethics clearance is current but it is not clear when it was initially sought. Is the ethics certificate an extension or only sought when the manuscript was submitted. In short, is ethics not sought before conduct of the research?

.

Reviewer #1: No

---

## [Author Response · Author response to Decision Letter 1]

21 Jan 2026

Yes, we have reviewed the comments and the response to editor and reviewers have been attached, thank you

---

## [Editor Report · Decision Letter 1]

3 Feb 2026

Dear Dr. Kalezi,

Thank you for submitting your manuscript to PLOS ONE. After careful consideration, we feel that it has merit but does not fully meet PLOS ONE’s publication criteria as it currently stands. Therefore, we invite you to submit a revised version of the manuscript that addresses the points raised during the review process.

We look forward to receiving your revised manuscript.

Kind regards,

Palesa Motshabi Chakane, PhD

Academic Editor

PLOS One

Journal Requirements:

Additional Editor Comments:

Dear Authors

The authors opined that "The goal is to assist healthcare providers and policymakers in improving care for these children.", however, there are no lessons to be drawn from this study. There is very limited data and result in just one table. There was no attempt to find any associations or inferrences to the outcomes reported. The Authors are therefore adviced to enhance the manuscript by

1. Reporting on echocardiographic detasils of cardiac lesions if any

2. Risk factors of adverse outcome

3. Further edit the manuscript fior consistency in reporting as shoen in the abstract

---

## [Author Response · Author response to Decision Letter 2]

19 Feb 2026

I have addresed the editors comments and attached the documents as suggested, thank you

---

## [Editor Report · Decision Letter 2]

5 Mar 2026

CLINICAL PROFILE AND HOSPITAL OUTCOMES OF CHILDREN WITH DOWN SYNDROME DIAGNOSED WITH CONGENITAL HEART DISEASE IN A DEVELOPING COUNTRY. A RETROSPECTIVE STUDY

PONE-D-25-46314R2

Dear Dr. Kalezi,

We’re pleased to inform you that your manuscript has been judged scientifically suitable for publication and will be formally accepted for publication once it meets all outstanding technical requirements.

Kind regards,

Palesa Motshabi Chakane, PhD

Academic Editor

PLOS One
---

## [Editor Report · Acceptance letter]

PONE-D-25-46314R2

PLOS One

Dear Dr. Kalezi,

I'm pleased to inform you that your manuscript has been deemed suitable for publication in PLOS One. Congratulations! Your manuscript is now being handed over to our production team.

Kind regards,

on behalf of

Dr. Palesa Motshabi Chakane

Academic Editor

PLOS One